# The Neonatal Fc Receptor Is Elevated in Monocyte-Derived Immune Cells in Pancreatic Cancer

**DOI:** 10.3390/ijms23137066

**Published:** 2022-06-25

**Authors:** Justin Thomas, Molly A. Torok, Kriti Agrawal, Timothy Pfau, Trang T. Vu, Justin Lyberger, Hsiaochi Chang, Alyssa Marie M. Castillo, Min Chen, Bryan Remaily, Kyeongmin Kim, Zhiliang Xie, Mary E. Dillhoff, Samuel K. Kulp, Gregory K. Behbehani, Zobeida Cruz-Monserrate, Latha P. Ganesan, Dwight H. Owen, Mitch A. Phelps, Christopher C. Coss, Thomas A. Mace

**Affiliations:** 1Division of Pharmaceutics and Pharmacology, College of Pharmacy, The Ohio State University, 496 W. 12th Ave., Columbus, OH 43210, USA; thomas.2962@osu.edu (J.T.); vu.179@osu.edu (T.T.V.); castillo.213@osu.edu (A.M.M.C.); chen.7274@osu.edu (M.C.); remaily.1@osu.edu (B.R.); kim.7648@osu.edu (K.K.); xie.53@osu.edu (Z.X.); kulp.1@osu.edu (S.K.K.); phelps.32@osu.edu (M.A.P.); 2The James Comprehensive Cancer Center, The Ohio State University, 496 W. 12th Ave., Columbus, OH 43210, USA; mollytorok17@gmail.com (M.A.T.); kriti17agrawal@gmail.com (K.A.); timothy.pfau@osumc.edu (T.P.); zobeida.cruz-monserrate@osumc.edu (Z.C.-M.); dwight.owen@osumc.edu (D.H.O.); 3Division of Hematology, Department of Internal Medicine, The Ohio State University, 420 W. 12th Ave., Columbus, OH 43210, USA; justin.lyberger@osumc.edu (J.L.); chang.763@osu.edu (H.C.); gregory.behbehani@osumc.edu (G.K.B.); 4Division of Surgical Oncology, Department of Internal Medicine, The Ohio State University, 420 W. 12th Ave., Columbus, OH 43210, USA; mary.dillhoff@osumc.edu; 5Division of Gastroenterology, Hepatology and Nutrition, Department of Internal Medicine, The Ohio State University, 420 W. 12th Ave., Columbus, OH 43210, USA; 6Division of Rheumatology and Immunology, Department of Internal Medicine, The Ohio State University, 420 W. 12th Ave., Columbus, OH 43210, USA; latha.ganesan@osumc.edu; 7Division of Medical Oncology, Department of Internal Medicine, The Ohio State University, 420 W. 12th Ave., Columbus, OH 43210, USA

**Keywords:** pancreatic cancer, neonatal Fc receptor, FcRn, tumor microenvironment, immunosuppression

## Abstract

The neonatal Fc receptor (FcRn) is responsible for recycling of IgG antibodies and albumin throughout the body. This mechanism has been exploited for pharmaceutic delivery across an array of diseases to either enhance or diminish this function. Monoclonal antibodies and albumin-bound nanoparticles are examples of FcRn-dependent anti-cancer therapeutics. Despite its importance in drug delivery, little is known about FcRn expression in circulating immune cells. Through time-of-flight mass cytometry (CyTOF) we were able to characterize FcRn expression in peripheral blood mononuclear cell (PBMC) populations of pancreatic ductal adenocarcinoma (PDAC) patients and non-cancer donors. Furthermore, we were able to replicate these findings in an orthotopic murine model of PDAC. Altogether, we found that in both patients and mice with PDAC, FcRn was elevated in migratory and resident classical dendritic cell type 2 (cDC2) as well as monocytic and granulocytic myeloid-derived suppressor cell (MDSC) populations compared to tumor-free controls. Furthermore, PBMCs from PDAC patients had elevated monocyte, dendritic cells and MDSCs relative to non-cancer donor PBMCs. Future investigations into FcRn activity may further elucidate possible mechanisms of poor efficacy of antibody immunotherapies in patients with PDAC.

## 1. Introduction

Fc receptors are a main regulator of humoral immune response. The many roles of Fc receptors include the phagocytic uptake of antibodies, activation of B cells, and maturation of dendritic cells [1]. Fc gamma receptors (FcγR) are a subset of cell surface Fc receptors with high affinity binding of immunoglobulin G (IgG) antibodies. Upon binding to IgG antibodies, FcγRs can propagate both activating and inhibitory immune signals while facilitating IgG endocytosis [2]. The neonatal Fc receptor (FcRn) is primarily intracellular, and functions as a heterodimer comprised of a major histocompatibility complex class I-like alpha chain and molecule of beta-2-microglobulin. Although originally discovered as the receptor responsible for maternal IgG transcytosis across the placental barrier [3], FcRn has since been implicated in multiple essential somatic functions. The best characterized include FcRn binding to both IgGs and albumin which prevents their catabolism following pino- or endocytosis in multiple cell types [4]. The ability of FcRn to salvage albumin and IgGs from lysosomal degradation affords an extended circulation half-life to these abundant serum proteins relative to other circulating factors of similar molecular mass [5]. FcRn binds client proteins with high affinity at low pH (~5.5–6) but has much lower affinity at physiological pH (7–7.4) [6]. Low binding at physiological pH is critical for client protein release back into the extra-cellular space and is essential for FcRn-dependent transcytosis. Within the gut epithelium, FcRn shuttles monomeric IgG from the lymph into the intestinal lumen, and IgG immune complexes back into the lymphatic system where they can activate dendritic cells and other professional antigen-presenting cells (APCs) [7].

Complex FcRn biology is exploited in multiple ways in therapeutic development, such as engineering Fc containing peptides [8], tuning mAb:FcRn interactions by engineering mAb Fc domains [9], and in creating albumin conjugated or albumin-bound therapy [10]. For example, Nab-paclitaxel is an albumin-bound nanoparticle that is commonly used in combination with other first-line therapies to treat various solid-tumor cancers including pancreatic ductal adenocarcinoma (PDAC). Since the nanoparticle is bound to albumin, its circulation is prolonged in the blood stream, improving the overall pharmacokinetic profile of paclitaxel relative to naked paclitaxel formulated in Cremophor [11]. Nab-paclitaxel’s improved pharmacokinetics are due, in part, to the recycling role of FcRn.

Several therapeutic mAbs have been specifically engineered to have enhanced binding affinity to FcRn to fine-tune their pharmacokinetic properties. Multiple single amino acid substitutions within the Fc domain can modulate FcRn’s pH dependent binding. For instance, mutations to Ile253, His310, and His345 that prevent protonation at low pH can completely ablate FcRn binding. Conversely, M252Y, S254T, and T256E substitutions can increase FcRn affinity at low pH, thus prolonging the circulating half-life of the mAb [12]. The elevated FcRn affinity strategy is used by the anti-IL6R antibody, tocilizumab, which has been shown to reduce tumor growth in mice with pancreatic ductal adenocarcinoma [13,14]. With mAb-based therapy growing in prevalence across multiple diseases, an improved understanding of the influence of patient FcRn on the biodistribution of Fc and albumin containing therapies is essential.

In addition to FcRn’s role in IgG and albumin recycling, multiple reports indicate FcRn is critical for proper antigen processing and presentation [15,16]. As a critical part of the adaptive immune system, antigen presentation encompasses the process by which antigen is engulfed, processed, and presented on the major histocompatibility complex by antigen-presenting cells (mainly dendritic cells, macrophages, and B cells) to activate a targeted T cell response [17]. Evidence supports FcRn’s role in the intracellular processing of antigen immune complexes, or antibody-opsonized antigens, such as neoantigens shed from tumors [15,16]. Notably, mice lacking dendritic cell FcRn exhibit reduced anti-tumor immunity and increased tumor burdens in spontaneous models of colorectal cancer [16]. While we appreciate that FcRn’s involvement in antigen presentation is still subject to heavy debate, the combined roles for FcRn in therapeutic disposition and immune response support an expanded understanding of this key protein’s expression and function in diverse cell types and disease states.

The expression and function of FcRn in endothelial cells is well characterized but there is surprisingly little known about FcRn within myeloid populations. Even basic information about FcRn expression within the three most common types of dendritic cell in humans has not been reported [18]. Macrophages and monocytes are two of the largest phagocytic/pinocytic cell populations in the body that are known to both express FcRn and provide large contributions to whole body IgG clearance [19]. However, to our knowledge, systemic changes in FcRn expression within these critical cell types in disease states have not been studied. Furthermore, studies on disease-related changes in FcRn that do exist focus exclusively on expression and activity intrinsic to disease tissue, such as tumor cells [20,21,22,23,24]. Taken together, these gaps in knowledge support a broader characterization of FcRn in systemic immune populations occurring as a function of disease.

PDAC is the third leading cause of cancer-related death in the United States with 5-year survival rates below 10%. PDAC is predicted to be the second most common form of cancer-related death by 2030 [25]. Only 20% of patients diagnosed with PDAC are eligible for surgery, with the majority receiving neoadjuvant chemotherapy with gemcitabine and nab-paclitaxel [26]. mAb-based immunotherapies have so far been ineffective in patients with PDAC which has been hypothesized to be due to the high local and systemic immunosuppression with low T cell tumor infiltration and low tumor mutation burden, leading to lower neoantigen potential [27,28]. Myeloid immune cell populations are highly elevated in PDAC patients correlating to poor survival and, recently, clinical trials are investigating how targeting these cellular populations can improve immunotherapy [29,30]. Immunosuppressive immune populations are elevated both systemically and locally within the TME which drive PDAC progression and growth.

Given PDAC’s broad resistance to mAb-based therapy [31], including immune checkpoint inhibitors [32], we used PDAC as a model disease to explore FcRn levels within circulating immune cells in tumor-bearing mice and cancer patients. PDAC tumors are well known to foster immunosuppressive environments, [33] making them an ideal system to investigate changes in circulating immune populations. Specifically, PDAC tumors have been described as having low T cell infiltration, low tumor mutational burden, and low neoantigen potential [27,28]. Given this context, we hypothesized that resistance to ICI and mAb-based therapies may be due in part to deleterious changes in FcRn expression within key circulating myeloid immune populations. In this study, we report significant changes in FcRn expression within dendritic cell, monocyte, and myeloid-derived suppressor cell populations in human and murine PDACs.

## 2. Results

### 2.1. PDAC Tumor-Bearing Mice Exhibit Altered FcRn Expression among Monocytic Immune Cell Populations

Given FcRn’s established role in mAb recycling and antigen processing in dendritic cells and macrophages, we sought to further explore FcRn expression in the different monocyte, DC, and MDSC populations within an orthotopic mouse model of PDAC (PDX-1-Cre, LSL-KrasG12D, LSL-Trp53^−/−^) [34]. Splenocytes from eight-week-old, C57BL6 tumor-free and tumor-bearing mice were analyzed by flow cytometry to determine percentages of various cell populations (Appendix A) as previously described [35]. Representative gating schemes and contour plots are shown (Figure 1A). As expected, we observed differences in immune populations when comparing pancreatic tumor-bearing mice to mice without tumors. Mice with pancreatic tumors had elevated percentages of gMDSC (CD11b^+^Ly6G^+^Ly6C^+/−^, *p* = 0.0005; Figure 1B), cDC1 (CD8^+^CD11b^−^CD103^−^CD24^+^, *p* = 0.0022, Figure 1C), migratory cDC2 (CD8^−^CD11b^+^CD103^+^CD24^++^, *p* = 0.0154; Figure 1D), and pDC (CD11b^−^CD11c^+^Ly6C^+^CD317^+^, *p* = 0.0079; Figure 1E), populations and reductions in Langerhans DC (MHC class II^+^CD11b^+^CD11c^+^Ly6C^−^DC-sign^+^, *p* = 0.0011; Figure 1F) populations compared to mice without tumors. No differences in macrophage (CD11b^+^CD64^+^F4/80^+^, *p* = 0.037; Figure 1G), moDC (MHC class II^+^CD11b^+^CD11c^+/−^Ly6C^+^DC-sign^+^; Figure 1H), mMDSC (CD11b^+^Ly6G^−^Ly6C^+^; Figure 1I), and cDC2 (CD8^−^CD11b^+^CD103^−^CD24^+/−^; Figure 1J) populations were observed between tumor-free and tumor-bearing mice.

Further analysis of these myeloid and APC populations revealed specific alterations of FcRn expression in mice with PDAC relative to tumor-free mice (Figure 2A). We characterized these changes in terms of FcRn positivity, or cells within that population that contain FcRn, as well as FcRn expression level by measuring mean fluorescence intensity (MFI). Figure 2B shows elevated FcRn positivity (*p* = 0.00014) and MFI (*p* = 0.017) of migratory cDC2 in tumor-bearing mice. mMDSCs also had significantly elevated FcRn positivity (*p* = 0.0023) and MFI (*p* = 0.00028) relative to tumor-free controls (Figure 2C). gMDSCs cells from tumor-bearing mice had a trend in increased FcRn positivity (*p* = 0.0542; Figure 2D) whereas cDC2s had significantly higher MFI (*p* = 0.036), but no change in positivity (Figure 2E). Langerhans DCs were the only cell type to show a near significant decrease (*p* = 0.0598) in FcRn positivity but no change in MFI (Figure 2F). There were no significant changes in FcRn positivity or MFI in moDCs (Figure 2H), pDCs (Figure 2I), cDC1 or macrophages (Figure 2J) between tumor-free and tumor-bearing mouse splenocytes (Figure 2G–J).

### 2.2. Immunophenotyping of Circulating PBMC from PDAC Patients Shows Alterations in Immune Populations

To probe the relationship between altered immune cell populations in tumor-bearing mice and patients, PBMCs from non-cancer obese patients (healthy control samples; *n* = 8) and PDAC patients prior to surgical resection (*n* = 13, Tumor Stage: 1–2) (Table 1) were subjected to CyTOF analyses. Antibody-bound PBMCs were gated into 41 different immune populations (Appendix A). Initial analyses focused on seven cellular immune populations, based on prevalence: T cells, B cells, Natural Killer Cells, Granulocytes, Monocytes, Dendritic Cell, and MDSCs (Appendix A). Representative gating is shown Figure 3A. T cells were significantly elevated in PDAC patients (*p* = 0.00092), with a median fold change increase of 1.75 (Figure 3B). Along with T cells, granulocytes were also significantly elevated in PDAC patients (*p* = 0.0078) with a median fold change of 3.4 relative to non-cancer controls, although they accounted for significantly less of the total immune population. In contrast, PDAC patient B cells were significantly decreased (*p* = 0.010) from 10% to 5% of the total immune population. The combined populations of granulocytic myeloid-derived suppressor cells (MDSCs) and monocytic MDSCs were also significantly decreased in PDAC patients (*p* = 0.043).

### 2.3. FcRn Is Localized to Myeloid-Derived Antigen-Presenting Cell Populations

We performed CyTOF combined with intracellular flow cytometry to assess expression of FcRn in immune populations in patients with PDAC compared to healthy control PBMC samples (Appendix A). Representative t-SNE plots of the most prevalent cell populations as well as FcRn localization mostly to dendritic cell and monocyte populations is shown (Figure 4A–B). These data show that FcRn is highly localized to myeloid and professional antigen presenting populations with elevated expression observed in individuals with PDAC compared to cancer-free individuals. While the prevalence of T cells and B cells were highly dependent on tumor status, their FcRn positivity were low and unchanged between non-cancer controls and PDAC patients, similar to natural killer cells (Figure 4C). Despite unchanging levels of total cells, granulocytes were shown to have significantly decreased number of FcRn positive cells from PDAC patients (*p* = 0.015). Monocyte (*p* = 0.030), dendritic cell (*p* = 0.025) and MDSC populations (*p* = 0.0027) with FcRn positivity were all markedly increased in PDAC patients relative to non-cancer patients. In non-cancer patients, these populations each had an FcRn positivity of ~75%, but significantly increased to almost 90% in PDAC patients. Monocytic and dendritic cell populations had no change in cell number between non-cancer and PDAC PBMCs. MDSCs were lower in number in PDAC patients, but their FcRn positivity was increased.

### 2.4. Increased Expression of FcRn in PDAC Patient Immune Populations

In addition to determining the number of cells that express FcRn, we also characterized the amount of FcRn expressed by each cell by quantifying the median metal intensity (MMI). Figure 5A is a normalized heatmap of each patient’s FcRn MMI within 18 different populations. Confirming observations from t-SNE analysis (Figure 4), T cell (*p* = 0.0018), and B cell (*p* = 0.0029) populations showed very low expression in all non-cancer and PDAC patients (MMI < 1.5 a.u. out of 34 a.u., Figure 5B–C). Granulocytes were the only population to show a significant decrease in FcRn MMI (*p* = 0.042) (Figure 5D). Also, the vast majority of FcRn expression was concentrated to monocyte (*p* = 0.017), DC (*p* = 0.017) and MDSC (*p* = 0.012) populations (Figure 5E–G). NK cells had no significant change between non-cancer and PDAC FcRn MMI (Figure 5H). Within the seven main cell populations previously described, FcRn expression was elevated in both monocyte and DC populations in PDAC relative to non-cancer PBMCs with granulocytes having little detectable FcRn expression. In conjunction with Figure 3 and Figure 4, these findings provide evidence that monocyte-derived immune cells have increased prevalence and intensity of FcRn expression in PDAC patients compared to non-cancer controls.

### 2.5. FcRn Is Localized to Monocyte-Derived Subpopulations

Classical monocytes (CD16^−^) accounted for over 90% of the total monocyte population, with nonclassical monocytes (CD16^+^) accounting for the balance of cells. In PDAC patients, the monocyte population trended towards the classical monocyte subpopulation (*p* = 0.061) and away from the non-classical monocyte subpopulation (*p* = 0.041) (Figure 6A). Classical monocyte FcRn positivity was also significantly increased from 59% to 71% (*p* = 0.038) (Figure 6B). FcRn MMI was elevated in classical (*p* = 0.026), but not non-classical monocytes (Figure 6C).

Dendritic cells were further divided into monocytic (CD11c^−^) and plasmacytoid (CD11c^+^) DC populations revealing pDCs accounted for less than 10% of total DCs as compared to 80–90% monocytic DCs (Figure 6D). Although there was no significant change in monocytic DC numbers, monocytic DC FcRn positivity was significantly increased in PDAC PBMCs (*p* = 0.034) (Figure 6E). Monocytic DCs also experienced elevated FcRn MMI in PDAC patients (*p* = 0.032), but pDCs showed no difference (Figure 6F).

MDSC populations were divided into monocyte-derived (CD66b^−^CD20^−^CD19^−^CD3^−^) and granulocyte-derived (CD66b^+^) MDSCs (Figure 6G). While the granulocytic MDSC subpopulation remained close to zero, the monocytic MDSCs were about 15% of the total cells stained and dropped nearly 3-fold within the PDAC PBMCs to 5% (*p* = 0.00018). Even though the total number of monocytic MDSCs decreased and the granulocytic MDSCs were low, both of their populations experienced significantly elevated FcRn positivity in PDAC patients (*p* = 0.034, *p* = 0.026, respectively) (Figure 6H). Monocytic MDSC FcRn MMI was significantly increased in PDAC PBMCs (*p* = 0.021), while granulocytic FcRn MMI had no change (Figure 6I). Together, these results further support the PDAC dependent induction of FcRn expression in highly specific cell populations of monocytic origin.

## 3. Materials and Methods

### 3.1. Murine Model of Pancreatic Cancer and Splenocyte Isolation

For murine orthotopic tumor studies, we used samples collected from a previously published study [34]. Briefly, 10^6^ syngeneic luciferase-expressing KPC tumor cells, originally derived from a primary pancreatic tumor in a KPC mouse (PDX-1-Cre, LSL-Kras^G12D^, LSL-Trp53^−/−^), were injected in Matrigel (BD Biosciences, Franklin Lakes, NJ, USA) into the tail of the pancreas of C57BL6/J mice. Bioluminescent imaging was used to monitor tumor growth, and tumor weight was measured at time of sacrifice. Three (3) weeks following orthotopic injections when mice had terminal tumor volume by imaging and weights, mice were euthanized, and spleens from tumor-free and tumor-bearing animals were removed and placed in PBS on ice. Under sterile conditions, the spleens were mashed with a 1 mL syringe plunger and strained twice through a 70 µm nylon cell strainer (Fisher Scientific, Waltham, MA, USA). The cells were centrifuged at 1700 rpm for 5 min before aspirating the supernatant. Splenocytes were resuspended in 10 mL red blood cell lysis buffer and pipetted to ensure total lysis of the red blood cells. The cells were resuspended in a freezing medium (90% FBS, 10% DMSO) and stored in liquid N_2_ prior to flow cytometry staining. All splenocytes collected from the prior study [34] were cryopreserved until use in this study.

### 3.2. Flow Cytometry Surface Marker Staining

Splenocytes were stained for the following immunocyte populations: monocytic dendritic cells (moDCs), classical dendritic cells (cDC), plasmacytoid dendritic cells (pDC), macrophages, and myeloid-derived suppressor cells (MDSCs). The cells were stained with the appropriate surface antibody markers for 30 min on ice. For flow cytometry intracellular staining of FcRn, splenocytes were permeabilized for 20 min on ice in the dark using 500 µL of the Cytofix/Cytoperm Buffer (BD Biosciences, Franklin Lakes, NJ, USA). Cells were stained using an anti-FcRn intracellular antibody (Biorbyt, Cat. No. orb360882, St. Louis, MO, USA) and analyzed on a Fortessa (BD Biosciences, Franklin Lakes, NJ, USA) flow cytometer. Antibodies used to stain for mouse antigens included: CD11c (Clone: N418; BioLegend, San Diego, CA, USA), MHC class II I-A/I-E (Clone: M5/114.15.2; BioLegend, San Diego, CA, USA), Ly6G (Clone: 1A8; BioLegend, San Diego, CA, USA), DC-Sign (Clone: MMD3; BioLegend, San Diego, CA, USA), CD11b (Clone: M1/70; BioLegend, San Diego, CA, USA), CD103 (Clone: 2E7; BioLegend, San Diego, CA, USA), CD8 (Clone: 53-6.7; Abcam, Cambridge, UK), CD24 (Clone: M1/69; BioLegend, San Diego, CA, USA), CD317 (Clone: 927; BioLegend, San Diego, CA, USA), F4/80 (Clone: BM8; BioLegend, San Diego, CA, USA), CD64 (Clone: X54-5/7.1; BioLegend, San Diego, CA, USA), and Ly6C (Clone: HK1.4; BioLegend, San Diego, CA, USA).

### 3.3. PBMC Staining and Mass Cytometry (CyTOF)

Peripheral blood mononuclear cells (PBMCs) were isolated from whole blood via density gradient centrifugation using Ficoll-Paque (Amersham, Pharmacia Biotech, Bjorkgatan, Sweden). Isolated cells were then surface stained with the Maxpar Cell Surface Stain Kit (Fluidigm, San Francisco, CA, USA) with the addition of CD11b (Clone: ICRF44; Fluidigm, San Francisco, CA, USA) and CD33 (Clone: WM53; BioLegend, San Diego, CA, USA) antibodies (Appendix A). Following the surface stain, the cells were fixed and permeabilized using the Intracellular Staining with True-Phos™ Perm Buffer in Cell Suspensions kit (BioLegend, San Diego, CA, USA). The cells were intracellularly stained for FcRn (Clone: 937508; R&D, which we conjugated to the 169-Tm metal isotope, Minneapolis, MN, USA). Cells were then washed twice with EDTA-enriched water and strained into filter-cap flow tubes with a magnetic bead acquisition solution. The solution was run through the Helios Mass Cytometer (Fluidigm, San Francisco, CA, USA) and the data was uploaded to Cytobank (Cytobank, Mountain View, CA, USA) for analysis. Cell population gating and heatmaps were created within Cytobank. T-distributed stochastic neighbor embedding (t-SNE) plots were also constructed in Cytobank.

### 3.4. Statistics

All statistics were performed in GraphPad Prism software (version 8.4.3 for Windows, San Diego, CA, USA). Welch’s *t*-tests were used to distinguish statistical significance of *p* < 0.05 between TF/PDAC or non-cancer/PDAC groups within each population. Statistical outliers were removed using the ROUT identify outlier test within GraphPad [36].

## 4. Discussion

Here we are the first to report elevated levels of FcRn in monocyte, dendritic cell and MDSC populations in both human and a murine model of PDAC (Table 2). However, the implications of this phenomenon remain unclear. PDAC has few presenting symptoms and is exceptionally lethal [37] such that it is commonly diagnosed after metastasis has occurred and treatment options are limited and largely ineffective [27,38]. Of particular interest are immune checkpoint inhibitors (ICIs), which re-activate host anti-tumor immune surveillance by targeting the axis of tumor, T cells and antigen-presenting cells [39]. Although there are currently nine FDA approved ICIs for multiple cancers, ICI therapy has so far been ineffective in PDAC [28,40]. Similarly, murine models of PDAC have limited responses to ICIs [34,41]. It has been hypothesized that this lack of efficacy may be due to the observations that PDAC patients are highly immunosuppressed with low T cell tumor infiltration and low tumor mutation burden, leading to lower neoantigen potential [27,28]. As previously noted, FcRn is integral to the processing and presentation of antigens within APCs [4,20,42,43]. If FcRn is unable to deliver its IgG payload to the appropriate destination, it cannot be properly processed and loaded onto the MHC for T cell presentation, and a dampened T cell response may result. It has already been reported that PDAC has low T cell tumor infiltration rates compared to other cancer types responsive to ICI therapeutics [44,45]. As such, reduced, not elevated, FcRn levels within APCs would be expected to trend with poor T cell infiltration and response due to low recognition of tumor neoantigens. Our conflicting finding of elevated FcRn invites further study into the mechanisms governing FcRn levels within APCs.

In addition to FcRn’s role in antigen presentation, its ability to traffic albumin bound or conjugated drugs may also be affected by changes in FcRn levels [46]. When bound to albumin, the pharmacologic properties of chemotherapeutics such as doxorubicin and paclitaxel can be improved along with therapeutic response [11,47]. Albumin-conjugated doxorubicin’s efficacy has been shown to be FcRn-sensitive in pancreatic cancers [10]. Albumin-bound paclitaxel (nab-paclitaxel, Abraxane^®^, Celgene Corporation, Summit, NJ, USA) is used as a first-line treatment of PDAC. Nab-paclitaxel is delivered in combination with gemcitabine and is regularly used as a neoadjuvant treatment for pre-surgical and metastatic PDAC patients [48]. Elevated FcRn may increase nab-paclitaxel internalization in tumor associated macrophages which was recently shown to be important for its anti-PDAC efficacy [49]. Future investigations of FcRn function will focus on its role in these monocytic immune populations and determine whether it directly mediates efficacy of IgG mAbs or albumin-bound chemotherapeutics.

The drastic upregulation of PDAC FcRn positivity/MFI in the murine cDC2 and migratory cDC2 cells (Figure 2B), and FcRn positivity and MMI in human classical monocyte populations (Figure 6A) (consisting of the classical dendritic cell populations) is particularly interesting. It is well understood that classical dendritic cell type 2 (CD11b^+^CD8^−^) have greater antigen presentation capabilities for CD4^+^ T cells than cDC1s [50], while cDC1s are better at cross-presentation to CD8^+^ T cells [51]. However, the cross-presentation induced by cDC2 cells is heavily dependent on FcRn expression [15]. Additionally, elevated FcRn expression in MDSCs (Figure 2C, Figure 4C, Figure 5G and Figure 6H–I) is intriguing. MDSC populations are known to expand in cancer, particularly around the tumor [52,53]. Since MDSCs are not professional antigen presenting cells and are not thought to play a role in IgG or albumin biodistribution, their abundant FcRn expression is puzzling. Further understanding of FcRn function in MDSCs is warranted. As suggested by a recent report [18], the antigen presentation and cross-presentation capabilities of cDC2s merit further study in light of the consistency of FcRn is upregulation in both murine PDAC cDC2s as well as human classical monocytes.

Some murine models of PDAC respond well to combination immunotherapies due to their mildly “inflamed” or T cell infiltrated tumor microenvironments [33]. However, clinical trials in PDAC patients have shown little to no response to ICI immunotherapies outside of MSI-high subsets [54,55]. Our data shows striking similarities between elevated human and murine FcRn expression and positivity in monocytes, dendritic cells and MDSCs in PDAC. The discrepancy in response between patients and model systems begs for a deeper examination into the functionality of Fc receptors within these populations to explore other mechanisms in the antigen presentation pathway that may differ between humans and mice. For instance, membrane-bound FcγRs are known to play a role in recruiting IgGs into the endosome-lysosome before they can bind to FcRn within the vesicle [56]. Humans and mice have distinct FcγRs with overlapping functions [57], suggesting species differences in FcγRs may play a role in disparate responses to immunotherapy.

We acknowledge that since the PBMCs came from surgery-eligible patients who had undergone neoadjuvant chemotherapy, their circulating immune populations do not reflect those with more severe disease or those that are treatment naïve. This study limitation is especially pertinent when considering MDSCs as they are particularly sensitive to chemotherapy and our data do not likely represent treatment-naïve patients’ MDSC populations. It is clear that there is a significant drop in MDSC cell number within the PDAC vs. non-cancer patients (Figure 3A), and while this does not properly reflect the natural circulating immune populations; it does show what a patient’s immune populations would be like during the course of treatments. Another limitation is the absence of immune phenotyping within the tumor microenvironment. However, results presented here on FcRn expression, given what is known about how circulating immune cells correlate with local immune cells and patient survival, help to bring more clarity to the immune landscape in this disease.

In this report, we are the first to demonstrate changes in FcRn expression within myeloid populations in patients and mice with pancreatic cancer. Our findings highlight disease-dependent changes in systemic FcRn levels in key cell populations governing both IgG biodistribution and immune response. Future investigations into changes in FcRn function that might accompany changes in FcRn levels are required to understand how our findings can help explain poor ICI response in PDAC patients. Our studies were limited to PDAC but support additional studies in other cancers, including those that are responsive to ICIs, as well as non-cancer diseases where mAb therapies are indicated. These studies are necessary to determine if disease-mediated changes in immune cell FcRn are widespread and will be the subject of future reports.

## Figures and Tables

**Figure 1 ijms-23-07066-f001:**
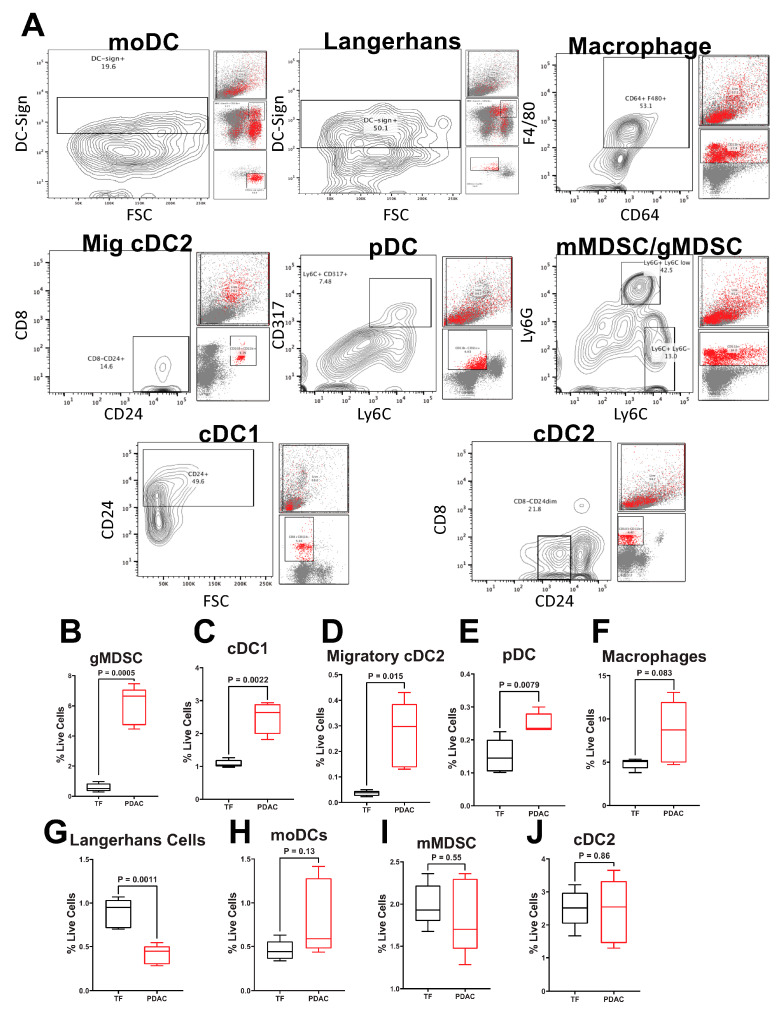
Pancreatic Cancer Alters Murine Splenocyte Immune Populations. Flow cytometry was performed on splenocytes from five different tumor-free (TF) or pancreatic tumor-bearing (PDX-1-Cre, LSL-Kras^G12D^, LSL-Trp53^−/−^) mice. All tumor-bearing mice were inoculated with tumor cells at the same time. (**A**) Representative contour plots gating for each of the nine populations in PDAC mice is shown. Back-gating of each population is provided to the right of each plot. Quantification of splenocyte live cell percentages of (**B**) gMDSC (CD11b^+^Ly6G^+^Ly6C^+/−^); (**C**) cDC1 (CD8^+^CD11b^−^CD103^−^CD24^+^); (**D**) migratory cDC2 (CD8^−^CD11b^+^CD103^+^CD24^++^); (**E**) pDC (CD11b^−^CD11c^+^Ly6C^+^CD317^+^); (**F**) Langerhans DC (MHC class II^+^CD11b^+^CD11c^+^Ly6C^−^DC-sign^+^; (**G**) macrophage (CD11b^+^CD64^+^F4/80^+^); (**H**) moDCs (MHC class II^+^CD11b^+^CD11c^+/−^Ly6C^+^DC-sign^+^); (**I**) mMDSC (CD11b^+^Ly6G^−^Ly6C^+^), and (**J**) cDC2 (CD8^−^CD11b^+^CD103^+^CD24^++^). Per animal percent live cells calculated by dividing the total number in each population by the total number of live cells stained. Biological replicates represent staining and analyses from five different animals per group. All samples analyzed were from a single experiment with five biological replicates per group. Individual biological replicates were combined into box and whisker plots representing the median and interquartile range (25–75%) with whiskers representing the outlying 25%. All statistical comparisons are unpaired t-tests with a Welch’s correction.

**Figure 2 ijms-23-07066-f002:**
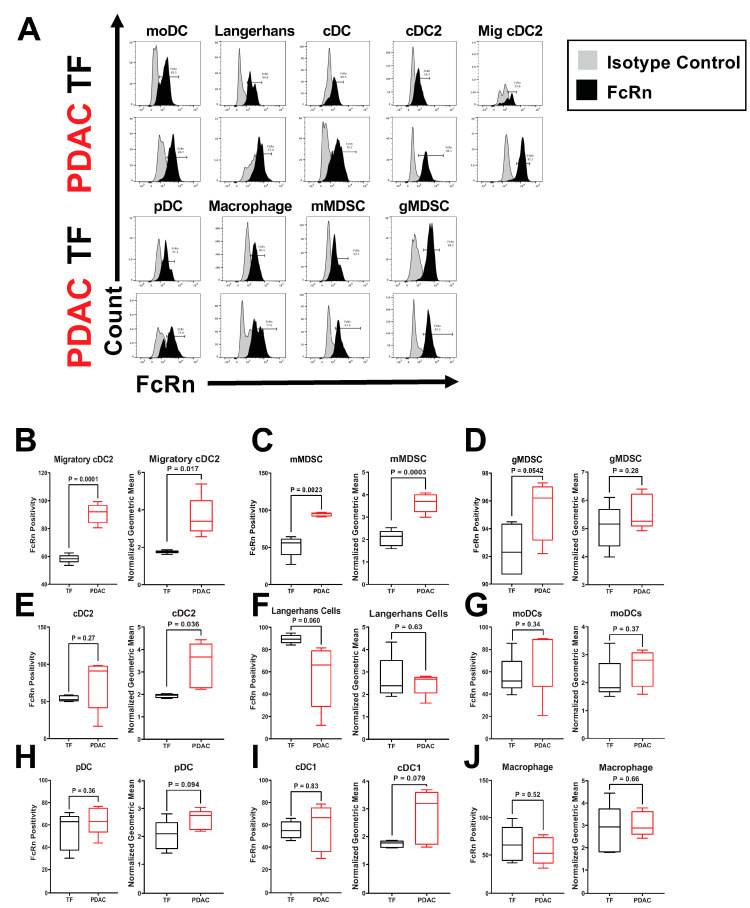
FcRn Positivity and Expression are Altered in Murine Splenocyte Immune Populations. Splenocytes from five different tumor-free (TF) or pancreatic tumor-bearing (PDX-1-Cre, LSLKras^G12D^, LSL-Trp53^−/−^) mice were permeabilized and stained with either anti-FcRn or an isotype control antibody. Biological replicates represent staining and analyses from five different animals per group. (**A**) Representative histograms of FcRn and isotype control are provided for TF and pancreatic tumor-bearing mice with isotype peaks in gray and FcRn peaks in black. Each cell type was assessed for FcRn positivity and the mean fluorescence intensity (MFI). Quantification of splenocyte FcRn positivity and mean fluorescence intensity (MFI) for (**B**) migratory cDC2 (CD8^−^CD11b^+^CD103^+^CD24^++^); (**C**) mMDSC (CD11b^+^Ly6G^−^Ly6C^+^); (**D**) gMDSC (CD11b^+^Ly6G^+^Ly6C^+/−^); (**E**) cDC2 (CD8^−^CD11b^+^CD103^+^CD24^++^), (**F**) Langerhans DCs (MHC class II^+^CD11b^+^CD11c^+^Ly6C^−^DC-sign^+^; (**G**) moDCs (MHC class II^+^CD11b^+^CD11c^+/−^Ly6C^+^DC-sign^+^); (**H**) pDC (CD11b^−^CD11c^+^Ly6C^+^CD317^+^); (**I**) cDC1 (CD8^+^CD11b^−^CD103^−^CD24^+^); and (**J**) macrophages (CD11b^+^CD64^+^F4/80^+^). Per sample FcRn positivity is measured by dividing the cells that express FcRn by the total cells within each population. Per sample normalized MFI is calculated by dividing the geometric mean of the FcRn channel for each sample by its isotype control. Biological replicates represent staining and analyses from five different animals per group. All samples analyzed were from a single experiment with five biological replicates per group. Individual biological replicates were combined into box and whisker plots representing the median and interquartile range (25–75%) with whiskers representing the outlying 25%. All statistical comparisons are unpaired t-tests with a Welch’s correction.

**Figure 3 ijms-23-07066-f003:**
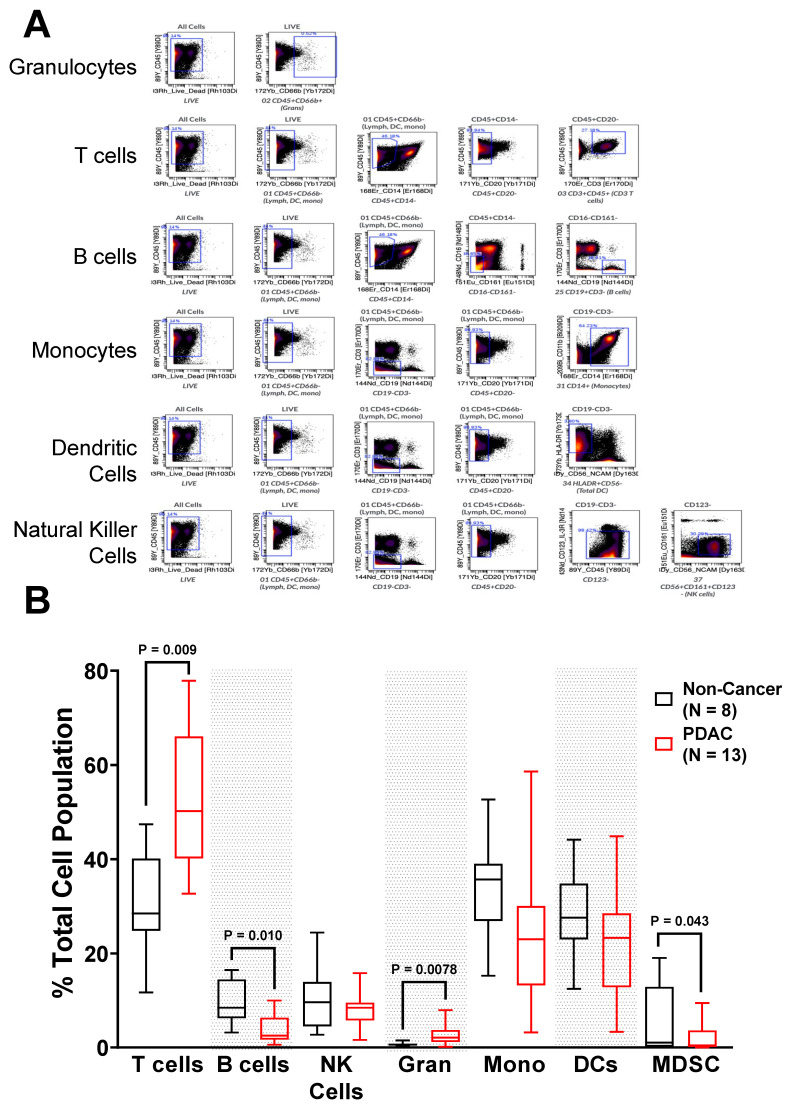
PDAC Patients Have Altered Circulating Immune Populations. Peripheral blood mononuclear cells (PBMCs) were collected and processed from non-cancer and PDAC patients, stained with isotope-bound antibodies, and analyzed by the mass cytometer. Biological replicates represent staining and analyses of *n* = 8 non-cancer and *n* = 13 PDAC patient samples. (**A**) Representative gating for granulocytes, T cells, B cells, monocytes, dendritic cells, and natural killer cells; and (**B**) Quantification of immune populations from non-cancer and PDAC patients. Granulocytes are denoted as Gran, and monocytes are denoted as Mono. Biological replicates represent staining and analyses of each individual sample. Individual biological replicates were combined into box and whisker plots representing the median and interquartile range (25–75%) with whiskers representing the outlying 25%. All statistical comparisons are unpaired *t*-tests with a Welch’s correction.

**Figure 4 ijms-23-07066-f004:**
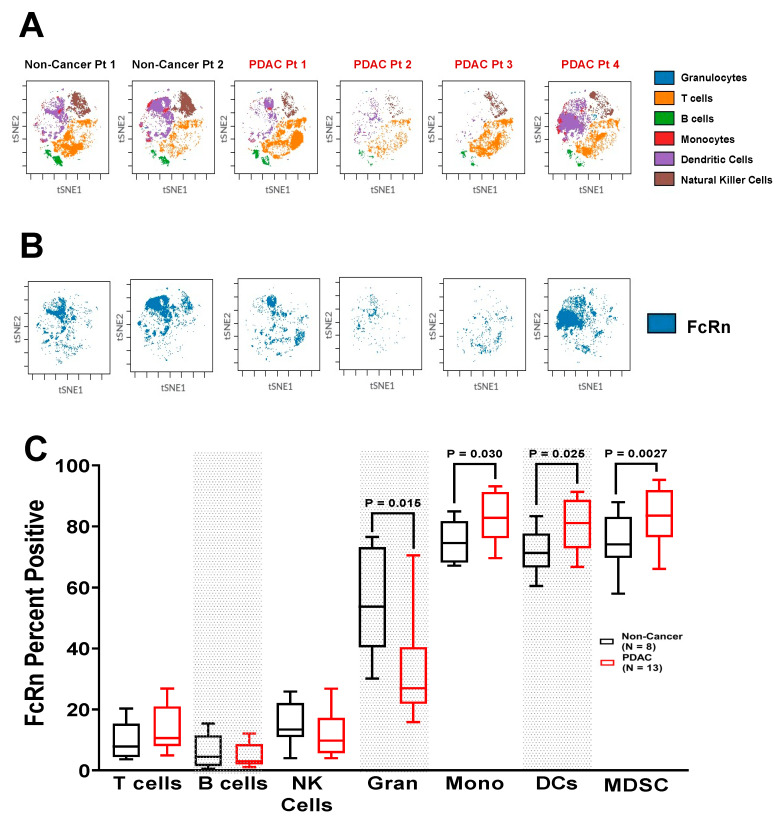
Circulating Myeloid Cells Expressing FcRn are Elevated in PDAC Patients. PBMCs of non-cancer (*n* = 8) and PDAC (*n* = 13) patients were permeabilized and stained for FcRn. Representative t-SNE plots of the (**A**) most prevalent cell populations; and (**B**) FcRn localization from two individual non-cancer and four individual PDAC patients are shown. (**C**) Quantification of FcRn expression within each immune population. Granulocytes are denoted as Gran, and monocytes are denoted as Mono. Biological replicates represent staining and analyses of each individual sample. Individual biological replicates were combined into box and whisker plots representing the median and interquartile range (25–75%) with whiskers representing the outlying 25%. All statistical comparisons are unpaired *t*-tests with a Welch’s correction.

**Figure 5 ijms-23-07066-f005:**
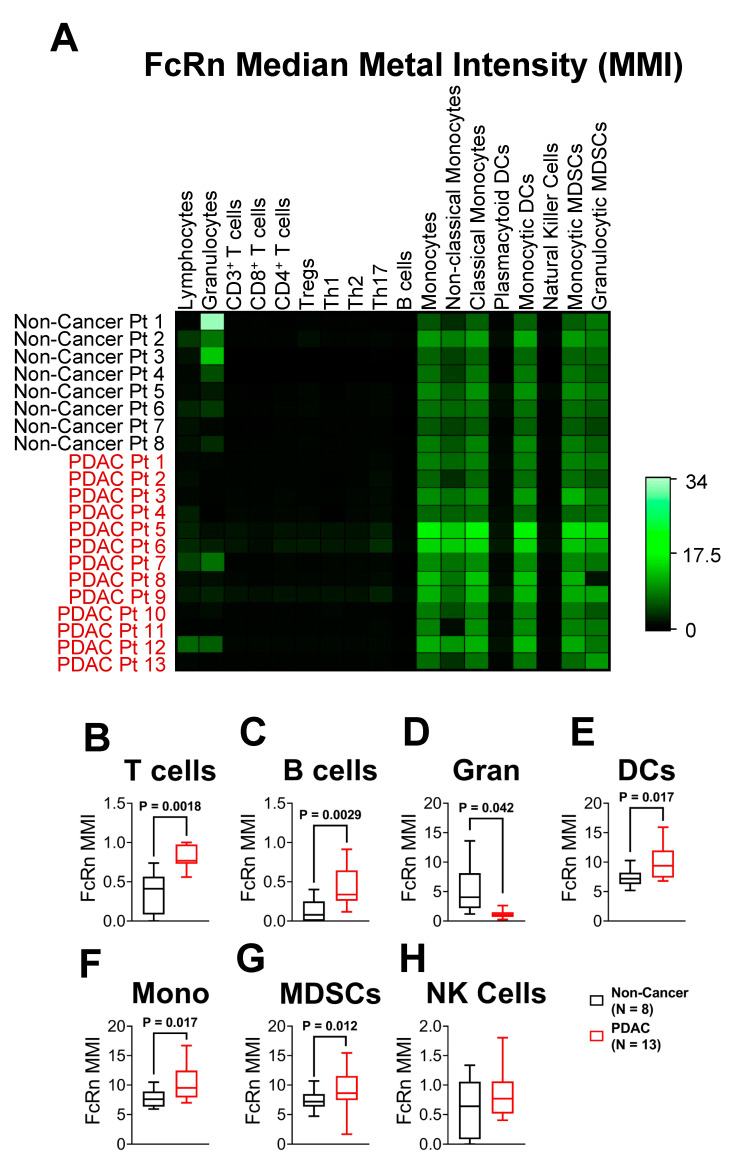
Myeloid FcRn Cellular Expression is Elevated in PDAC Patients. Per-cell expression of FcRn quantified by analyzing the mean metal intensity (MMI; non-cancer, *n* = 8; PDAC, *n* = 13) (**A**) All MMI values were normalized and provided in a representative heatmap for each individual sample. In addition to immune populations from Figure 3A, subsets of T cells, monocytes, dendritic cells and MDSCs are presented. Quantified MMI plots of non-cancer (*n* = 8, black) and PDAC (*n* = 13, red) (**B**) T cells, (**C**) B cells, (**D**) granulocytes, (**E**) dendritic cells (**F**) monocytes, (**G**) MDSCs, and (**H**) NK cells. Box and whisker plots represent the median and interquartile range (25–75%) with whiskers representing the outlying 25%. All statistical comparisons are unpaired *t*-tests with a Welch’s correction.

**Figure 6 ijms-23-07066-f006:**
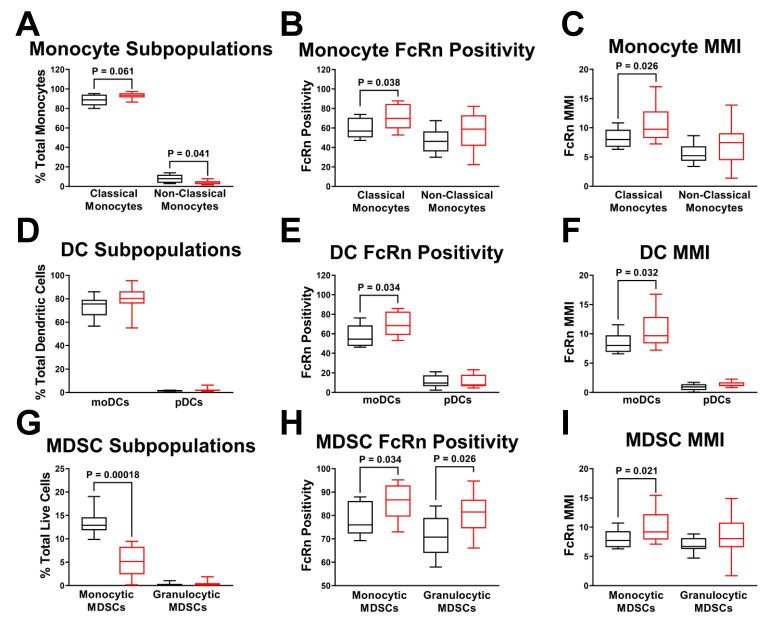
Myeloid Cell Subpopulations have Differential Expression of FcRn. Subpopulations of monocytes, dendritic cells and MDSCs (**A**) Monocytes subdivided into classical and non-classical status with quantification of (**B**) monocyte FcRn positivity and (**C**) monocyte FcRn MMI (**D**) Dendritic cells subdivided into monocytic dendritic cells and plasmacytoid dendritic cells with quantification of (**E**) DC FcRn positivity and (**F**) DC FcRn MMI. (**G**) MDSCs were subdivided into monocytic and granulocytic MDSC populations with quantification of (**H**) MDSCs FcRn positivity and (**I**) MDSC FcRn MMI. Non-cancer patient (black, *n* = 8) and PDAC patient (red, *n* = 13). Box and whisker plots represent the median and interquartile range (25–75%) with whiskers representing the outlying 25%. All statistics are done with unpaired *t*-tests with a Welch’s correction.

**Table 1 ijms-23-07066-t001:** PBMC Donor Demographic Information. The demographic information for PDAC and non-cancer controls.

Sex		PDAC	Non-Cancer
	Male	6	5
	Female	7	3
Race			
	White	11	N/A
	Black/African American	1	N/A
Initial Diagnosis			
	PDAC	10	N/A
	Pancreatic Mass	3	N/A
Neoadjuvant Treatment			
	Gemcitabine/Abraxane	1	N/A
	FOLFIRINOX	9	N/A
Height (in.)	Median (Range)	65.98 (58.4–72.6)	65.47 (61.0–72.8)
Weight (lbs.)	Median (Range)	152.9 (107.58–198.1)	221.01 (173.94–261.25)
BMI	Median (Range)	25.14 (17.38–32.39)	32.57 (30.29–43.28)
Serum Pre-albumin	Median (Range)	18.5 (9–29)	N/A
Serum Albumin	Median (Range)	3.8 (3.3–4.4)	N/A
Serum Glucose	Median (Range)	105 (87–207)	88.5 (76–100)

**Table 2 ijms-23-07066-t002:** Summary Table of FcRn Positivity and MFI/MMI. Summary table for the changes in monocytic DC, plasmacytoid DC, classical DC, macrophage, gMDSC and mMDSC FcRn positivity and MFI/MMI in both mouse and human samples relative to the tumor-free (TF) controls.

Cell Type	Mouse FcRn Positivity	Human FcRn Positivity	Mouse FcRn MFI	Human FcRn MMI
Monocytic DC	No Change	Elevated	No Change	Elevated
Plasmacytoid DC	No Change	No Change	No Change	No Change
Classical DC	No Change	Elevated	Elevated	Elevated
Macrophages	No Change	No Change	No Change	No Change
Granulocytic MDSC	No Change	Elevated	No Change	No Change
Monocytic MDSC	Elevated	Elevated	Elevated	Elevated

## Data Availability

The data that support the findings of this study are available from the corresponding author upon reasonable request.

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
