# Peer review of "The Neonatal Fc Receptor Is Elevated in Monocyte-Derived Immune Cells in Pancreatic Cancer"

_ijms, 2022, doi:10.3390/ijms23137066_

Round 1

Reviewer 1 Report

This manuscript by Thomas and colleagues is well-written and valuable. But there are concerns as follows:

INTRODUCTION: This section needs to be improved to make it more specific and set the stage for why this study was performed.

1. Can we contextualize the major alterations in the local and systemic immune system among PDAC patients. 

2. The section on FCRn can be more truncated and made specific to cancer physiology. 

3. Can the authors describe a study hypothesis and the significance of the study rather than mere "exploring" FCRn in PDAC patients.

Re: Study significance: The authors describe poor ICI response among PDAC patients. Is the study designed for the final aim to discover targets to improve ICI responsiveness?

some points have been described in the discussion. they need to be introduced early on!

METHODS:

Well-written!

Appreciate the mouse model. compliant with ARRIVE guidelines:https://arriveguidelines.org/sites/arrive/files/documents/ARRIVE%20Compliance%20Questionnaire.pdf

4. The PDAC patients are not treatment-naive. So the inference from a treated patient is a study limitation later on.

5. Do you have matched PDAC tissue samples and the analyses of the local PDAC immune environment will be quite valuable. I wonder about the difference in local and systemic immune milieu? The results from this can be described later on. If not available, please address in the discussion as a limitation or available evidence on the correlation of PMBC with the local PDAC immune environment.

6. Statistics: please check if T-test can be used in non-paramteric data. Table 1: Please provide stage information of Pancreatic cancer. 

DISCUSSION:

7. Please clarify if the immune cells being investigated are from treatment-naive vs patients receiving Chemotherapy. How would prior treatment have impacted the current result?

8. The section on antigen-presenting is very well-written and should be expanded.

Author Response

We thank the reviewer for their time and their insightful comments and concerns. Below are our responses to their specific inquiries.

INTRODUCTION: This section needs to be improved to make it more specific and set the stage for why this study was performed. 

Can we contextualize the major alterations in the local and systemic immune system among PDAC patients. 

Thank you for this comment. We agree, including a brief discussion about local and systemic immune cells in PDAC improves the manuscript and have updated our introduction accordingly.

The section on FCRn can be more truncated and made specific to cancer physiology.  

Thank you to the reviewer for suggesting a streamlined discussion of FcRn better fit the narrative of the manuscript. We have removed some of the non-immune and/or IgG-transfer functions of FcRn from the introduction as suggested.

Can the authors describe a study hypothesis and the significance of the study rather than mere "exploring" FCRn in PDAC patients.

Thank you for suggesting a more explicit declaration of our hypothesis.  We updated our manuscript accordingly.

Re: Study significance: The authors describe poor ICI response among PDAC patients. Is the study designed for the final aim to discover targets to improve ICI responsiveness?

Thank you for this observation.  We were not explicitly looking for drug targets to improve ICI response in PDAC patients.  Given FcRn’s importance in mAb disposition and antigen presentation we wanted to evaluate potential changes in its expression in PDAC patients as a potential explanation for the failures of these therapies in these patients.

some points have been described in the discussion. they need to be introduced early on!

METHODS:

Well-written!

Appreciate the mouse model. compliant with ARRIVE

We thank the reviewer and have included the completed ARRIVE document to be compliant with guidelines.

The PDAC patients are not treatment-naive. So the inference from a treated patient is a study limitation later on.

We fully agree with the reviewer that inclusion of chemo-treated patients is a study limitation. We have added this into the discussion to clearly highlight this.

Do you have matched PDAC tissue samples and the analyses of the local PDAC immune environment will be quite valuable. I wonder about the difference in local and systemic immune milieu? The results from this can be described later on. If not available, please address in the discussion as a limitation or available evidence on the correlation of PMBC with the local PDAC immune environment. 

We thank you for the intriguing comparison and agree that these data would strengthen our study. Unfortunately, we do not have access to these samples. As suggested, we have added this to the discussion as a limitation of the study.

Statistics: please check if T-test can be used in non-parametric data. 

We apologize for the confusion, but we are unsure which data the reviewer is referring to as non-parametric.  Our statistical test, Welch’s T test, is more conservative than student’s T tests as it does not assume equal variance and should be appropriate for the comparisons made in the manuscript.

Table 1: Please provide stage information of Pancreatic cancer. 

Unfortunately, we do not have access to detailed tumor staging information for our patients. However, as all of the PBMCs were from patients who were about to undergo surgical resection, they were all Stage 1-2. We updated our results section to capture this important detail.

DISCUSSION:

Please clarify if the immune cells being investigated are from treatment-naive vs patients receiving Chemotherapy. How would prior treatment have impacted the current result?

Mdsc levels lower on pts, all neo chemo tx

We thank the reviewer for raising this concern.  We have included a paragraph acknowledging this study limitation in the discussion.

The section on antigen-presenting is very well-written and should be expanded.

We thank the review for the kind words about our discussion of antigen presentation but are unsure which part of our manuscript the review wants expanded as antigen presentation is discussed throughout.

Reviewer 2 Report

The article by Thomas et al. entitled “The neonatal Fc receptor is elevated in monocyte-derived immune cells in pancreatic cancer,” is a well-conducted, pre-clinical study evaluating a potential mechanism for poor efficacy of antibody immunotherapies in patients with pancreatic ductal adenocarcinoma (PDAC). The finding report FcRn was elevated in migratory and resident cDC2 as well as monocytic and granulocytic MDSC populations compared to tumor-free controls. They also demonstrate changes in FcRn expression within myeloid populations in patients and murine model with pancreatic cancer. 

In general, this is a highly significant topic as little known about the mechanism of FcRn-dependent anti-cancer therapeutics. The study is well-designed and the findings hold novelty. I have some suggestions to strengthen the study.

1. There is a question that lacking age information of patients in PBMC donor demographic information. Whether the patients are all at the same stage of age for comparison
?Do age differences regulate FcRn expression and distribution in immune populations?

2. Whether the malignancy of the tumor is positively correlated with the expression of FcRn in immune population in patients with PBMC sample? 

3. In this paper, the author mainly analyzes the immune cell population in PBMCs from non-cancer obese patients and PDAC patients.  If PBMCs from patients with chronic pancreatitis could be compared again, this could lead to a more precise analysis.

4. The p-values should be labeled on each set of histograms (like in Figure 5). This shows the differences between groups more clearly.

5. In the abstract, when the abbreviation "MDSC" appears for the first time, it should be clearly explained.

Author Response

There is a question that lacking age information of patients in PBMC donor demographic information. Whether the patients are all at the same stage of age for comparison?Do age differences regulate FcRn expression and distribution in immune populations?

We thank the reviewer for this insightful comment. Unfortunately, we do not have access to patient age information for the samples in our cohort.  As we expand our analyses in future studies, we are capturing rich patient/donor data including age etc.

Whether the malignancy of the tumor is positively correlated with the expression of FcRn in immune population in patients with PBMC sample? 

We agree that an improved understanding of the relationship between tumor burden and changes in host FcRn expression is essential.  As with the demographic information, we sadly do not have access to tumor staging or tumor burden information in from our patient samples.  We are starting to analyze samples from a cohort of metastatic PDAC patients via CyTOF and future analyses could examine FcRn expression as it correlates to cancer progression and overall survival.

In this paper, the author mainly analyzes the immune cell population in PBMCs from non-cancer obese patients and PDAC patients.  If PBMCs from patients with chronic pancreatitis could be compared again, this could lead to a more precise analysis.

We agree, given the increased rates of PDAC in patients with chronic pancreatitis, that these patients would be an exceptional control for non-cancerous pancreatic disease with which to study changes in host FcRn expression.  Regrettably, we do not have access to PBMCs from pancreatitis patients at this time.

The p-values should be labeled on each set of histograms (like in Figure 5). This shows the differences between groups more clearly.

Thank you for this suggestion. We replaced the asterisks in each figure with p values, and modified figure legends accordingly to highlight the differences between groups more clearly.

In the abstract, when the abbreviation "MDSC" appears for the first time, it should be clearly explained.

We apologize for this oversight.  We clarified each abbreviation in the abstract including cDC2 and MDSC.

Reviewer 3 Report

In this study, authors reported an increased expression of neonatal Fc receptor (FcRn) in monocyte-derived immune cells in PDAC. This study is interesting and well performed. However, the quality of the figure should be improved. A few other minors are suggested.

Amplify Figure 1 A, Figure 2A, and Figure 3 A are too small to see the number.

Figure 4, reduce the size of 4C and increase the sizes of 4A and 4B.

Abbreviations, cDC2 in the abstract, PDAC in lines 64, 83,

Number and unit, such as line 139, 500μL.

Author Response

Amplify Figure 1 A, Figure 2A, and Figure 3 A are too small to see the number.

We thank the reviewer for this comment.  We updated the manuscript to include a higher resolution image and bigger copies of each of these figure to improve readability.

Figure 4, reduce the size of 4C and increase the sizes of 4A and 4B.

We thank the reviewer for this suggestion to improve the readability of figure 4.  We changed the sizes accordingly in our updated manuscript to highlight the t-SNE plots in Figure 4.

Abbreviations, cDC2 in the abstract, PDAC in lines 64, 83,

In our updated manuscript we defined all abbreviations at their first use including those in the abstract.

Number and unit, such as line 139, 500μL.

We thank the reviewer for pointing out this error, we updated line 139 and confirmed all other similar errors were corrected.